# Sarcopenia in Children with Solid Organ Tumors: An Instrumental Era

**DOI:** 10.3390/cells11081278

**Published:** 2022-04-09

**Authors:** Annika Ritz, Eberhard Lurz, Michael Berger

**Affiliations:** 1Department of Pediatric Surgery, Dr. von Hauner Children’s Hospital, Ludwig-Maximilians-University Munich, 80337 Munich, Germany; annika.ritz@med.uni-muenchen.de; 2Department of Pediatrics, Division of Gastroenterology and Hepatology, Dr. von Hauner Children’s Hospital, Ludwig-Maximilians-University Munich, 80337 Munich, Germany; eberhard.lurz@med.uni-muenchen.de; 3Department of General, Abdominal, and Transplant Surgery, Essen University Hospital, 45147 Essen, Germany

**Keywords:** sarcopenia, psoas muscle surface area, children, biomarker

## Abstract

Sarcopenia has recently been studied in both adults and children and was found to be a prognostic marker for adverse outcome in a variety of patient groups. Our research showed that sarcopenia is a relevant marker in predicting outcome in children with solid organ tumors, such as hepatoblastoma and neuroblastoma. This was especially true in very ill, high-risk groups. Children with cancer have a higher likelihood of ongoing loss of skeletal muscle mass due to a mismatch in energy intake and expenditure. Additionally, the effects of cancer therapy, hormonal alterations, chronic inflammation, multi-organ dysfunction, and a hypermetabolic state all contribute to a loss of skeletal muscle mass. Sarcopenia seems to be able to pinpoint this waste to a high degree in a new and objective way, making it an additional tool in predicting and improving outcome in children. This article focuses on the current state of sarcopenia in children with solid organ tumors. It details the pathophysiological mechanisms behind sarcopenia, highlighting the technical features of the available methods for measuring muscle mass, strength, and function, including artificial intelligence (AI)-based techniques. It also reviews the latest research on sarcopenia in children, focusing on children with solid organ tumors.

## 1. Definitions of Sarcopenia

Sarcopenia is a syndrome named from the Greek words sark-, sárx, meaning “flesh”, and penía, meaning “poverty.” Despite its popularity as a research topic in the past decade, sarcopenia remains a subject of great debate. In 2016, the European Society for Clinical Nutrition and Metabolism (ESPEN) defined this syndrome as a progressive and generalized loss of skeletal muscle mass, strength, and function (performance) with a consequent risk of adverse outcomes [1]. The European Working Group on Sarcopenia in Older People (EWGSOP) updated its own definition of sarcopenia in 2018 (EWGSOP2). Previously defined as a progressive and generalized skeletal muscle disorder associated with an increased likelihood of adverse outcomes, the new definition puts low muscle strength at the forefront [2].

Sarcopenia can be categorized into primary and secondary causes. Primary causes include the primary breakdown in muscle as a result of aging, while secondary causes include disuse (low physical activity or immobility), side effects of medication, endocrine processes, inadequate intake of energy or protein, and disease-related mechanisms (e.g., neurodegenerative disease, cancer, organ failure, etc.) [1]. While historically, sarcopenia was associated with older adults, a reduction in muscle mass, strength, and function due to chronic disease has also been found in much younger individuals.

## 2. Changes in Muscle and Body Composition during Childhood

The molecular mechanisms of skeletal muscle homeostasis are complex. Put simply, muscle atrophy occurs when protein breakdown outweighs protein synthesis. One of the major pathways for protein synthesis is the insulin-like growth factor 1 (IGF-1)/Akt/mammalian target of rapamycin (mTOR) pathway, which is activated by various upstream factors, including IGF-1, insulin, and amino acids [3]. IGF-1 and insulin regulate the activation of phosphatidylinositol-3-kinase (PI3K), which in turn activates protein kinase B (Akt) and mTOR [4]. In cancer cachexia, the major signaling pathways that may contribute to protein breakdown are the IkB kinase (IKK)-nuclear factor (NF)-kB pathway, Akt-forkhead box O (FoxO) pathway, Janus kinase (JAK)/signal, Activator of transcription (STAT) pathway, and Transforming growth factor-beta (TGF-β)/Myostatin-Smad2/3 pathway [5].

Muscle protein is regularly broken down and synthesized during the course of a day. This turnover is highest during infancy and decreases thereafter [6]. During infancy, males have a greater fat-free mass (FFM), total body water, and total body potassium compared with females [7]. It is essential to mention that FFM is not the same as skeletal muscle mass (SMM) or lean mass (LM), as FFM not only includes muscle but also water, bone, and non-osseous material. On the other hand, LM is comprised of FFM and essential fats (Figure 1). During the first two years of life, the skeletal muscle component of FFM increases, and total body water and extracellular water decrease [7]. During the first four months of age, infants have an increase in adipose tissue, with percent body fat peaking to about 30% at around three to six months [7,8].

During childhood, muscle fiber size increases, while muscle fiber satellite cell content decreases relative to muscle fiber area [12]. Males have twice the number of muscle cells compared with females. In addition, males have 150% of the lean body mass of the average female [13].

During puberty, the increase in growth promoters, such as growth hormone, insulin, and IGF-1, may contribute to muscle growth [14,15]. In prepubertal children ages 3 to 14 years with a growth hormone deficiency, treatment with human growth hormone significantly increased muscle tissue and decreased adipose tissue [16]. Sex hormones also influence changes in muscle mass during puberty—male rats given testosterone or dihydrotestosterone during late puberty show an increase in muscle mass [17]. In comparison, female sex hormones during puberty may decrease muscle mass. After ovariectomy, female rats show an increased rate of skeletal muscle protein synthesis, while rats after estradiol or progesterone replacement have a lower synthesis rate [18]. Females gain more fat mass (FM) during puberty, while males gain more lean body mass [19]. After puberty, a balance is reached between muscle protein breakdown and synthesis [6].

## 3. Causes of Sarcopenia in Children with Solid Organ Tumors

Solid organ tumors make up about 30% of tumors in the pediatric population. The most common solid organ tumors in children include brain tumors, neuroblastoma, rhabdomyosarcoma, Wilms’ tumor, and osteosarcoma [20]. Causes of sarcopenia in patients with cancer have primarily been studied in adults, with fewer studies conducted in the pediatric population and even fewer directly targeting infants, children, or adolescents with solid organ tumors. Below is a summary of possible factors involved in reducing muscle mass, strength, and function in this population (Table 1).

### 3.1. Disuse

In children and adolescents with cancer, fatigue has been reported as one of the most prevalent symptoms [21,22]. Fatigue in this population has also been reported to impact participation in regular activities [21]. Although cancer-related fatigue in children and adolescents improves over time, in a subset of childhood cancer survivors, fatigue and low activity levels continue to persist years after treatment [23,24,25]. In adults, immobilization has been shown to reduce muscle mass and muscle fiber size [26,27]. After only ten days of bed rest, healthy adults were found to have a significant loss of lower extremity strength and power [28]. It is plausible that reduced activity in pediatric cancer patients contributes to sarcopenia. Currently, studies isolating the effects of immobility on muscle mass and strength in healthy children are not available. This is likely since immobilization mainly occurs in children who are ill.

### 3.2. Medications and Cancer Treatment

Chemotherapy treatment may, directly and indirectly, cause sarcopenia. Certain chemotherapy drugs, such as cisplatin and doxorubicin, have been found to directly activate the transcription factor NF-kB, causing muscle degradation in mice [29,30,31]. Moreover, chemotherapy stimulates oxidative stress, which increases reactive oxygen species (ROS) in the muscle [32]. Chemotherapy also increases TGF-β proteins, upregulating myostatin and pushing the muscle towards catabolism [33,34].

Indirectly, chemotherapy can cause fatigue, reducing physical activity. In addition, due to their quick turnover, cells in the gastrointestinal tract are damaged by chemotherapy [35]. Children consequently experience diarrhea, nausea, vomiting, and mucositis, causing a limited oral intake, reduced nutrient uptake, and fluid loss [36]. Cancer treatment can also lead to decreased food intake by causing changes in appetite and taste [37,38].

It is challenging to differentiate chemotherapy’s direct and indirect factors in children with solid organ tumors receiving treatment. This is in part because most children receive a multi-drug treatment regimen. Nakamura et al. studied changes in muscle mass during chemotherapy in children with high-risk neuroblastoma, measuring the psoas muscle index (PMI) after the first and third chemotherapy cycle and at the end of induction chemotherapy. Children were treated with cisplatin, cyclophosphamide, vincristine, and pirarubicin. Findings showed the most substantial reduction in PMI occurs at the beginning of chemotherapy treatment. Additionally, some subgroups (younger children and boys) tended to show recovery in muscle mass between the second and last measurements [39].

Another factor influencing muscle mass is radiation therapy. In young adult rats receiving ≥5 Gy of radiation, satellite cell numbers were reduced by 70%. The same study found reduced nitric oxide levels, which may partly explain the inhibition of satellite cell proliferation after radiation [40]. Radiation is also thought to cause breaks in the DNA of satellite cells. With low doses of radiation, satellite cell activation and proliferation are inhibited. Higher doses of radiation cause cell death, muscle necrosis, and muscle atrophy [41,42]. Radiation exposure also has long-term effects. At a median of 20 years after radiation exposure, 80% of children between 3.5 and 10 years of age with prior sarcomas of the extremities were found to have atrophy of the muscle and soft tissue [43].

Steroids may also be administered to children during the treatment of solid organ tumors. Especially in children who receive solid organ transplantation, immunosuppressive drugs, such as steroids, are a mainstay. In children with brain tumors, including metastases, steroids play a role in managing edema and associated neurological deficits [44]. Children with opsoclonus-myoclonus-ataxia syndrome, a condition associated with neuroblastoma, may benefit from treatment with prednisone [45,46]. Steroids can cause muscle protein degradation via the ubiquitin–proteasome and autophagy–lysosome systems. Steroids also suppress the synthesis of new muscle protein [47]. By decreasing the production of IGF-1 and by increasing myostatin production, steroids inhibit the activation of satellite cells and inhibit proliferation and differentiation of myoblasts, causing muscle atrophy [48]. In growing rats receiving steroids, a decrease in the rate of protein synthesis and a transient increase in protein degradation have been observed [49]. Likewise, in children ages 6 to 17 years receiving oral prednisone (2 mg/kg/day with a maximum of 60 mg/d) after an initial diagnosis of Crohn’s disease, a significant whole-body protein breakdown and loss was observed after 14 days [50]. It is important to note that muscle atrophy primarily occurs when steroids are given continuously or at high dosages, while intermittent steroid use may promote muscle synthesis [51].

### 3.3. Endocrine Processes

While insulin, growth hormone, IGF-1, and testosterone have anabolic effects, glucagon, glucocorticoids, and catecholamines have catabolic effects. In patients with cancer, proinflammatory cytokines such as interleukin (IL)-1, IL-6, tumor necrosis factor alpha (TNF-α), interferon-alpha (IFN-α), and interferon-gamma (IFN-γ) are released by the tumor or the host’s immune system in reaction to the tumor [52]. These proinflammatory cytokines activate the hypothalamic–pituitary–adrenal axis, which leads to the production of cortisol in the adrenal glands. In the adrenal gland and central nervous system, activation of the sympathetic nervous system causes deregulated production of catecholamines norepinephrine and epinephrine [53]. These stress hormones generate resistance to insulin and growth factors, thus inhibiting muscle growth [54].

Furthermore, evidence suggests that the peptide ghrelin, which is produced in the stomach and stimulates appetite, plays a role in muscle wasting. In adult patients with neuroendocrine [55], gastric [56,57], and lung tumors [58], ghrelin was considerably elevated in the plasma. This elevation may suggest that the body is attempting to counteract anorexia, which is often seen in cancer patients—an endocrine response to the ‘ghrelin resistance’ observed in cancer patients—or a reaction to the muscle breakdown associated with cancer [3,59,60]. Ghrelin directly inhibits proteolysis by blocking protein degradation caused by cytokines and by stopping muscle cell apoptosis induced by the chemotherapy agent doxorubicin [3,61]. In addition, resistance to the hormones Neuropeptide Y (NPY) and Agouti-related protein (AgRP), which promote food intake, has also been observed in tumor-bearing animals [62,63]. In pediatric patients with cancer, total ghrelin levels have only been studied in children with leukemia. In this study, children with ALL had significantly lower serum levels of ghrelin compared to healthy controls. This difference was more pronounced pre-treatment and improved over time, suggesting an impact of ghrelin on cancer cachexia in children [64].

While initially discovered in intestinal cells, vitamin D receptors (VDRs) are also expressed in the nucleus of muscle cells and play essential roles in muscle function. When VDRs are inhibited in the muscle, a decrease in the rapid intracellular entry of calcium is observed, and muscle contractility is affected [65]. Micro- and macronutrient deficiency due to chemotherapy could be one reason why vitamin D deficiencies may be present in patients with cancer. Although the ideal concentration of Vitamin D in the blood and the treatment of vitamin D deficiencies are still widely debated topics, a recent study showed that adults with colorectal cancer with higher vitamin D levels had lower mortality [66]. In a heterogeneous group of children aged 0 to 18 years with solid organ tumors prior to cancer treatment, their vitamin D levels were lower compared with controls. Additionally, pre-treatment vitamin D levels correlated with prognosis [67].

### 3.4. Malnutrition and Cachexia

Malnutrition is defined as a state in which a lack of intake or uptake of nutrition causes body composition and body cell mass to become altered. In turn, this leads to a decrease in physical and mental function and a worse clinical outcome [1]. The etiology of malnutrition can be categorized into malnutrition without disease, disease-related malnutrition (DRM) without inflammation, and DRM with inflammation [1]. About 50 to 80% of adult patients with cancer experience chronic disease-related malnutrition with inflammation, also known as cancer cachexia [1,3]. In a cohort of children ages 5 to 16 years with mixed tumors—mainly hematological tumors, but also solid tumors—45% were found to be malnourished [68]. Fearon et al. described cancer cachexia as a “multifactorial syndrome defined by an ongoing loss of skeletal muscle mass (with or without loss of fat mass) that cannot be fully reversed by conventional nutritional support and leads to progressive functional impairment. Its pathophysiology is characterized by a negative protein and energy balance driven by a variable combination of reduced food intake and abnormal metabolism” [69]. No consensus exists on the definitions of malnutrition and cancer cachexia for children. While both sarcopenia and cachexia involve a loss of muscle mass, their etiologies vary. As described above, sarcopenia has many causes. While cachexia can lead to secondary sarcopenia, sarcopenia does not lead to malnutrition or cachexia. In progressive disease, cachexia can cause both a reduction in fat and muscle mass [70]. Since cachexia and sarcopenia frequently co-occur and no consensus exists on how to diagnose the two conditions, they can be challenging to differentiate.

Cancer cachexia is an energy balance disorder where energy intake is reduced and/or energy expenditure increases. Furthermore, in cancer cachexia, the body’s inflammatory response is activated. When proinflammatory cytokines (e.g., IL1-α, IL1-β, IL-6, IL-8, TNF-α, and IFN-γ) are released—mainly by immune cells—it leads to STAT3 phosphorylation and downstream myostatin activation, as described above. Cytokines are also transported across the blood–brain barrier and suppress appetite [37,71]. One direct example is TNF-α, which may deter food consumption by causing the patient to experience a bitter taste [72]. In patients with cancer, energy intake may also be reduced because of decreased appetite, chemotherapy-induced side effects, tumor obstruction of the gut, and psychological factors [3].

Energy deficits might also occur in children with solid organ tumors due to tumor processes [73], multi-organ dysfunction [3], pain stemming from invasive tumors [74], as well as stress during hospitalization [75]. Patients continuously struggle to maintain energy homeostasis as the tumor consumes glucose and amino acids, especially glutamine, for growth [73]. Tumor growth is also fueled by degrading protein in the muscle. Glutamine is directly taken up by the tumor and used as a nitrogen donor to synthesize protein and deoxyribonucleic acid (DNA). Simultaneously alanine is used to generate glucose in the liver, which is transported to the tumor for energy. In addition, the lactic acid produced in the tumor is recycled back to the liver through the Cori cycle at a high energy cost [3,37,76,77]. Studies have also suggested that cancer causes a decrease in mitochondrial adenosine triphosphate (ATP) synthesis in skeletal muscle and impairment of mitochondrial function (decreased oxidative capacity [78], disrupted protein synthesis [79], changes in membrane fluidity [79], and oxidatively modified mitochondrial proteins [80]), leading to calcium deregulation and ultimately muscle wasting [3].

Not only is energy wasted in the liver by utilizing the Cori cycle to feed the tumor, but it is also suggested that cancer decreases the efficiency of oxidative phosphorylation in the liver mitochondria [81]. Additionally, the liver also releases pro-inflammatory factors and secretes acute-phase proteins [82]. Lastly, it has been suggested that the liver plays a role in downregulating the circulation of very-low-density lipoproteins, increasing the expression of TSC22 domain family member 4 (TSC22D4), inhibiting lipogenesis, causing hepatic steatosis, and potentially stimulating liver gluconeogenesis, further increasing energy expenditure [83,84].

Cardiac dysfunction is frequently observed in patients with cancer and tumor-bearing animals [85,86]. Eleven days after syngeneic sarcomas were implanted into adult mice, significant cardiac atrophy and reduced cardiac myofibrillar, collagen, and soluble proteins were found [87]. Further studies in mice found that cancer cachexia may bring about cardiac atrophy [88,89]. Late-onset fibrosis [90] and altered composition of contractile proteins (e.g., troponin I and myosin heavy chain) [85] have also been observed in patients with cancer. Additionally, an increase in heart oxygen consumption, possibly due to anemia, and an increased heart rate contribute to energetic inefficiency [91] and, in the long term, cardiac dysfunction.

Recently, researchers have found that the gut also plays a role in activating the inflammatory response in patients with cancer, although our understanding is still in its infancy. Low levels of lactobacilli and high levels of Enterobacteriaceae have been discovered in mice with cancer cachexia [92,93,94]. When lactobacilli were given to mice with cancer cachexia, muscle atrophy was reversed [92,95]. Enterobacteriaceae are pro-inflammatory. Microbe-associated molecular patterns (MAMPs), such as the endotoxin lipopolysaccharide (LPS), are found on the bacterial surface. When disruption of the gut barrier occurs, Enterobacteriaceae and LPS enter the circulation. LPS then binds to Toll-like receptor 4 (TLR4) on host immune cells, triggering the transcription of proinflammatory cytokines (TNF-α, IL-1β, and IL-6) [96,97,98]. Interestingly, IL-6 has been shown to increase tight junction permeability. Anti-IL-6 antibodies prevented the induction of markers of muscle atrophy and reduced changes in gut barrier function [99]. Although cancer treatments contribute to mucositis and gut barrier dysfunction, increased gut permeability was also present in cachectic mice with leukemia receiving no cancer treatment [93]. Similar results have been found in adults with acute myeloid leukemia [100]. More research is needed to determine whether these findings apply to humans with cancer cachexia, specifically children with solid organ tumors.

## 4. Sarcopenic Obesity

The term sarcopenic obesity (SO) was coined in adults to describe a condition in which sarcopenia occurs together with an increase in FM [2,101]. SO has been observed in adult patients with cancer and has been associated with a higher risk of chemotherapy toxicity, surgical complications, and reduced survival rates [102,103,104]. Research on SO and its effects in pediatric patients with cancer is extremely limited, the pathophysiology of SO in this patient group is also not well understood, and its definition has not been agreed upon [101]. In the pediatric population, SO has been most commonly studied in children with acute lymphocytic leukemia (ALL). The findings show a frequent occurrence of SO during treatment and even years after survival [105,106,107,108]. Children with hematologic malignancies had more significant increases in fat mass than children with solid organ tumors [68]. In a heterogenous group of children with malignancies, 26% of which had solid organ tumors, a significant amount had an increased FM and decreased FFM during cancer treatment [109]. Conversely, another study in children with solid organ tumors or hematologic malignancy undergoing treatment found only increased fat mass in both groups [68]. The consequences of SO in children with cancer have not been determined.

## 5. Frailty

First described by Fried and colleagues in older adults, frailty is defined as an individual’s increased vulnerability and decreased capacity for resisting to a health stressor as well as an individual’s decreased adaptive capacity for regaining their initial clinical status afterward. It is characterized by five qualities, namely slowness, weakness, shrinkage, exhaustion, and diminished physical activity [110,111]. Both malnutrition and sarcopenia can lead to frailty. Ultimately, frailty is associated with increased mortality in several medical scenarios [1,111]. Using the Fried criteria, frailty was first described in children with chronic liver disease in a North American multicenter study using established pediatric tests for each quality. The results showed that a cut-off score of greater than five best defined frailty. Children with end-stage liver disease had significantly higher scores than children with compensated chronic liver disease, with weakness and slowness contributing most heavily to the score [112]. In the second available study on frailty in the pediatric population, children with complex cardiac defects were found to have lower z-scores for each quality compared with healthy controls [113]. Total frailty scores were not calculated in that study. The downside of the frailty assessment in children is that children need to be at least five years old to perform the respective tests. Given the major impact of muscle function on frailty scores in children with ESLD, frailty might be a valid clinical test to follow up with sarcopenia and might replace psoas muscle imaging in older children [112,113].

## 6. Diagnosing Sarcopenia

Currently, no standardized method exists to diagnose sarcopenia. In children, muscle mass has been measured using methods such as anthropometrics [114]; ultrasound [115]; bio-electric impedance analysis (BIA) [116]; dual-energy x-ray absorptiometry (DXA) [117]; computed tomography (CT) [118,119]; magnetic resonance imaging (MRI) [120]; and occasionally, densitometry [121], deuterium dilution [122], creatinine excretion [123], total body potassium [124], and neutron activation [125]. Not only are there a wide range of modalities used to determine muscle mass but also, within each modality, there is a lack of consensus on the definition of sarcopenia. Caution must be used when using measurements of FFM to determine SMM, as the breakdown of FFM changes during childhood. In children with solid organ tumors, DXA, CT, and MRI are most commonly used to determine muscle mass. Even among these modalities, markers (e.g., total psoas muscle area (tPMA) vs. SMM; total body lean mass vs. appendicular lean mass), equations (e.g., for muscle indices), and reference values vary (Table 2). Without a consensus on definitions for sarcopenia, over- and underestimations are likely.

### 6.1. Modalities Used to Measure Muscle Mass

The least invasive and expensive method that has been used to estimate skeletal muscle mass is anthropometrics, including the mid-upper arm circumference (MUAC), triceps skinfold thickness (TSF), and calf circumference (CC). Previous studies in adults have shown that the MUAC correlates with FFM, while the TSF correlates with FM on BIA and DXA [131,132]. Arm muscle area (AMA) is calculated using both the MUAC and TSF, with equations using the AMA in adults predicting skeletal muscle mass [133]. That being said, while the CC has been shown to predict performance and survival in adults, the EWGSOP2 does not recommend using anthropometrics to measure muscle mass [2]. A study examining anthropometrics and body composition in children showed MUAC and AMA were poor predictors of FFM [134]. The limitations of using the MUAC and TSF are inaccuracy in patients with edema, hypoalbuminemia, and obesity [135,136]. Another established anthropometric marker is body mass index (BMI). However, BMI is not a suitable marker for FM and FFM in children with cancer. When FFM decreases and FM increases, nutritional deficits may be overlooked when using BMI [137]. In patients 1 to 21 years old with solid organ tumors (Wilms tumor, Ewing sarcoma, osteosarcoma, or rhabdomyosarcoma), no association was found between BMI and change in body composition (skeletal muscle, visceral adipose tissue, subcutaneous adipose tissue, intermuscular adipose tissue, and residual lean tissue) on CT imaging between diagnosis and the first follow-up visit [128].

Another method that can be used to asses skeletal muscle in children is ultrasound. Using ultrasound, real-time measurements of muscle thickness and cross-sectional area of muscles can be performed [138]. While an ultrasound is portable, noninvasive, and does not exposure patients to radiation, it can be challenging to differentiate muscle from fat, fluid overload may influence muscle thickness, errors in measurements can occur when excessive pressure is applied to the muscle, and it is not a suitable technique for children under three years of age due to its low sensitivity [139,140]. Reference values for measuring muscle thickness and muscle force of the rectus femoris and vastus intermedius muscles measured mid-thigh have been published for children [115]. In children, the reliability of markers, such as the quadriceps femoris muscle thickness, and their usefulness as a measure of muscle mass and function need to be further examined.

Skeletal muscle mass can also be estimated using BIA and respective prediction equations [141]. While inexpensive, safe, and easy to use, BIA also has limitations. BIA body composition measurements are sensitive to hydration status, recent physical activity, and time spent lying down. Especially in children with liver, kidney, or cardiac disease, where fluid overload is a problem, measurements have a high risk for error. Overall, muscle mass estimations show a large individual prediction error [142]. In adults with liver cirrhosis, using the bioimpedance phase angle rather than bioimpedance body compartments has been shown to correlate with muscle mass and may be a better technique to identify low muscle mass in certain populations [143,144,145]. For children and adolescents 5 to 18.8 years of age, age- and gender-specific reference curves for appendicular (limb) skeletal muscle mass (SMMa) using BIA have been published [116]. For younger children, age-, gender-, and ethnicity-specific prediction equations are still needed. Rather than using bioimpedance body compartments, indices such as the bioimpedance phase angle have been shown to correlate with muscle mass in adults with liver cirrhosis. 

Air displacement plethysmography (ADP) is a non-imaging method that has previously been used to predict FM and FFM. This method measures changes in pressures within two chambers to evaluate volume. While ADP is non-invasive, calculations can be influenced by body temperature and moisture [146]. Using this technique, Murphy-Alford et al. found that a heterogeneous group of cancer survivors, 28 percent of whom were children previously diagnosed with solid organ tumors, had significantly increased fat mass. Contributing factors included poor diets, low physical activity, and excessive screen time [147].

DXA has become a widely used method for diagnosing sarcopenia in adults and children due to its low radiation exposure, short scan time, low costs, and relative ease of use. However, the limitations of DXA include an inaccuracy in patients with edema [148], an inaccuracy in patients with obesity (underestimates FFM) [149], and an inability to quantify fatty infiltration of muscle [142]. In addition, DXA results have been found to vary according to the instrument used [142]. Initially, DXA equations estimating SMM using appendicular lean tissue mass (ALM) were derived from adult equations. While adult predictive equations applied to adolescents with at least Tanner stage 5, equations no longer correlated with SMM in younger children. As a result, new equations were developed [150]. The accuracy of SMM predictions has improved with new equations accounting for weight, height, and ethnicity [151]. Currently, various groups have published age- and gender-specific reference values beginning with children who are three years old [150,151,152,153,154,155]. With reference values established for children, overexposure to radiation can be avoided.

CT and MRI scans are the gold standards for measuring skeletal muscle mass in adults [2]. While highly accurate [156], CT images have a high cost and cause radiation exposure. With MRI scans, accuracy in measuring skeletal muscle remains high with the added benefit of no radiation exposure, but they come with a high price and lengthy acquisition time [157]. That being said, in certain patient populations (e.g., children with solid organ tumors), CT and MRI scans are often readily available. In addition, research in adults has shown that CT and MRI images can be used interchangeably to determine skeletal muscle mass and quality [158]. Similar results have been found when comparing subcutaneous and visceral adipose tissue between CT and MRI images in adults and adolescents [159,160]. Reference values for subcutaneous and visceral adipose tissue using MRI imagining have also been published for children 6 to 18 years of age [161].

### 6.2. Measuring Muscle Mass on CT and MRI

Among studies using CT and MRI to evaluate muscle mass, measurement fields (e.g., whole-body vs. single-slice on various lumbar levels) vary. Most CT and MRI studies performed evaluated one or more markers using a single-slice measurement. Multiple studies have found that single-slice measurements of the abdominal skeletal muscle area (SMA) are accurate predictors of whole-body SMM [9,162,163]. Using both CT and MRI, this correlation has been demonstrated in adults using single-slice measurements on and between lumbar levels L3–L5, with lower lumbar levels showing stronger correlations. Similar studies have not been conducted in the pediatric population. The mid-thigh cross-sectional muscle area is also a good predictor of whole-body SMM [164], but not many studies have been published at the mid-thigh level, possibly because—unlike abdominal scans—these images are not readily available in many patient groups. Muscle markers to quantify muscle mass, such as the tPMA, axial psoas muscle thickness, paraspinal muscle area, dorsal muscle group area, and the abdominal wall muscle area as well as muscle quality markers such as intramuscular adipose tissue content and muscle attenuation have been used to evaluate for sarcopenia [165,166]. As muscle mass is dependent on body size, some studies assessing sarcopenia opt to adjust measurements, e.g., by dividing by squared height, weight, or BMI. In both adults and children, no clear recommendation exists for whether adjustments should be made for body size [2]. It is important to consider that, during puberty, weight is proportional to height cubed rather than height squared [167,168].

The SMA and the tPMA, as well as their indices (used to adjust for height), are the two common markers used to assess skeletal muscle loss on CT and MRI imaging in children with solid organ tumors. The tPMA is calculated by adding the left and right psoas muscle areas (PMAs) on a single-slice image [118,127,169,170,171,172,173,174,175]. Both the SMA and the tPMA have been found to predict outcome in adults and children with cancer [119,176,177,178]. Recently, age- and sex-specific reference values for the PMA have been published in adults and children [179,180,181,182]. However, it is important to note that some authors argue the PMA does not represent the total skeletal muscle mass due to its relatively small size [183,184].

Body composition and muscle cross-sectional areas can be measured using region of interest tools or semi-automated software programs, such as sliceOmatic (TomoVision, Montreal, QC, Canada), ImageJ (National Institutes of Health, Bethesda, MD, USA), FatSeg (Biomedical Imaging Group Rotterdam of Erasmus MC, Rotterdam, The Netherlands, using MeVisLab (Mevis Medical Solutions, Bremen, Germany)) and OsiriX (Pixmeo, Geneva, Switzerland) [185]. One study showed excellent levels of agreement for cross-sectional muscle area, visceral adipose tissue area, and subcutaneous adipose tissue area on adult abdominal CT scans among all programs [186]. However, a second study showed a significant difference in skeletal muscle index (SMI), FM, FFM, and mean skeletal muscle Hounsfield Units between SliceOmatic and OsiriX, concluding that although a clinically significant difference is doubtful, using the same software package for serial measurements is recommended [187]. As new programs and technologies become available, researchers must make a choice about whether to incorporate these programs into their methodology. Multiple studies assessing sarcopenia in children with solid organ tumors have implemented semi-automated software programs, primarily sliceOmatic [128,129,130,188]. While the psoas muscle has relatively sharp borders on CT and MRI images, especially at lower lumbar levels, measuring the SMA on a cross-sectional image can be more difficult. This may be why studies measuring the SMA in children with solid organ tumors tend to use semi-automated programs.

In the past few years, research has been conducted to fully automate single-slice muscle mass measurements using deep learning models. Using fully convolutional neural network-based algorithms, among others, the measurement slice is detected and muscle mass areas are segmented [189,190,191]. Radiomics, another machine learning approach, has also been used to assess muscle mass by extracting radiomic features from CT images [192]. While more research is needed to test these new automated machine learning algorithms in larger patient cohorts and patients with various diseases, in the next few years, evaluating for sarcopenia may be transformed by more easily and quickly identifying patients at risk for poor outcomes. 

### 6.3. Measuring Muscle Strength and Function

Another important component of sarcopenia is muscle strength and function. In adults, muscle grip strength and the chair-stand test are two common ways to assess muscle strength in the clinical setting. Function (physical performance) is evaluated by measuring gait speed, a 400 m walk, and completing the timed-up-and-go test or short physical performance battery protocol [2]. Many studies on sarcopenia in children focus on muscle mass rather than strength or function. Currently, standardized protocols for these measurements (e.g., handgrip strength and six-minute walk test) have only been established for children older than three to four years [193,194,195,196,197]. In healthy children ages 5 to 11 years, thigh muscle size strongly correlated with strength [198].

Especially in young children, testing for muscle strength and function may have to be adapted as factors such as motivation, the ability to follow instructions, and developmental maturity may impact the results of testing. Research has shown that the handgrip strength test can be accurately used in children as young as four and a half years old [199]. Not only is the handgrip strength test a valuable predictor of total muscle strength in children and adolescents but also handgrip strength relative to body mass index (grip-to-BMI ratio) may be a valuable tool for predicting the presence of SO in children [200,201]. Using a hand-held dynamometer and measuring isometric muscle force, researchers obtained measurements with fair to excellent reliability in children as young as two years of age [202]. For school-aged children, push-ups, pull-ups, and the standing long jump test may also give further insight into muscle strength. Created to assess musculoskeletal fitness in U.S. schools, reference curves have been established for handgrip strength and modified pull-ups in healthy children as young as three years [203]. Reference values for muscle strength testing to assess pediatric physical fitness have also been published for specific ethnic populations and in various countries [204,205]. To evaluate muscle function, the six-minute walk test has been used in children as young as three years [196]. In 2002, the American Thoracic Society published guidelines for the six-minute walk test, and since then, many studies from different countries have published reference values for the test [206,207]. Another test used to assess muscle function in children is the timed up and down stairs test [208]. 

Testing for muscle strength and function becomes more challenging in infants and toddlers. In infants, clinical maneuvers to test for muscle strength include “pulling to sit”, “shoulder suspension”, and “ventral suspension”. Not only is it difficult to identify muscle weakness and hypotonia, it is similarly difficult to quantify that muscle weakness once identified [209]. In 2012, researchers developed a new measurement method to quantify muscle strength in infants and toddlers starting at six months of age by employing a pulling task to evoke a maximal pulling action [210]. A recent study in adults has also found that the cross-sectional area of the mid-thigh region on CT correlates with muscle strength, while the CT attenuation value correlated with physical performance [211]. Further research is needed to determine if inferences about strength and function can be made using available CT and MRI scans in the pediatric population. 

## 7. Current Research on Sarcopenia

In recent years, sarcopenia has become a popular research topic. The prevalence, consequences, and treatment of sarcopenia in older adults, as well as sarcopenia’s ability to predict outcome in adults with chronic illnesses and those undergoing surgery, are frequently examined topics. However, not only are studies difficult to compare due to the different modalities and muscle markers used but also many studies have varying definitions and cut-offs for sarcopenia. Sarcopenia has been associated with a worse outcome in adults with gastrointestinal cancer, breast cancer, lung cancer, esophageal cancer, pancreatic cancer, ovarian cancer, non-hematologic solid tumors, colorectal cancer, and head and neck cancer, to name a few [176,177,178,212,213,214,215,216,217,218,219,220].

Limited research has been published on sarcopenia in the pediatric population, although interest has increased in the last few years. Multiple studies have been published on how sarcopenia relates to children with end-stage liver disease (ESLD) and liver transplantation (LT). Mangus et al. demonstrated significant sarcopenia, determined by scaling the PMA on CT by height, in patients with end-stage liver, kidney, and intestine failure [221]. Measuring the tPMA at L3–4 and L4–5 on CT imagining, Lurz et al. determined that children with ESLD prior to LT had a lower tPMA than healthy age- and gender-matched controls [118]. In children undergoing LT, sarcopenia and lower muscle markers have been associated with a higher risk of surgical complications, longer hospital stays, longer pediatric intensive care unit (PICU) stays, higher re-operation rates, and higher risk for mortality [222,223,224,225]. Mager et al. retrospectively determined that persistent sarcopenia on DXA scans after LT in children with ESLD is associated with poorer growth and recurrent hospital admissions [226]. A low muscle mass has also been found in newly diagnosed children with Crohn’s disease and vitamin D deficiency [227], in hospitalized children diagnosed with complex appendicitis [172], in children with autoimmune liver disease [228], and in children with inflammatory bowel disease [229]. In children with ulcerative colitis, a low paraspinous muscle area was associated with poor outcome after colectomy [230].

In the subcategory of pediatric oncology, muscle mass and sarcopenia have most frequently been studied in ALL. Rayar et al. found that children with ALL show a reduction in SMM on DXA early in treatment, with the degree of reduction determining the burden of illness [231]. Similarly, Suzuki et al. measured the PMA before and after induction therapy in children with ALL and found all patients experience muscle loss. Adverse events during treatment were more commonly seen in sarcopenic patients [232]. In the long-term, children who survive ALL have a risk of developing SO, thereby negatively impacting their health-related quality of life [106].

### Sarcopenia in Children with Solid Organ Tumors

Much of the research on sarcopenia in children with solid organ tumors has been studied in children with neuroblastoma (NB), with many of the studies published in 2021. Kawakubo et al. measured the rate of change in the tPMA on CT imaging at lumbar height L3 in children with high-risk NB before and after receiving standard treatment. While the rate of change in the tPMA increased in the progression-free survival group to 1.24, it decreased to 0.84 in the relapse and death group, showing that a delta in tPMA greater than 1.00 may be a useful predictor for prolonged overall and progression-free survival [127]. Although reference values for pediatric patients have recently been published, regional differences in reference values may exist. Using the rate of change as a marker to determine the individual change in muscle mass may overcome this obstacle. The limitations of this study include its small sample size of only 13 patients. Additionally, it is important to consider that the standard treatment protocol for NB can vary according to the region. Thus, the results may vary when other protocols for high-risk NB are used.

Similarly, Nakamura et al. and Ijmpa et al. also took longitudinal approaches when measuring muscle mass. Nakamura et al. measured muscle mass on CT imagining at diagnosis, after the first and third chemotherapy cycle, and at the end of induction chemotherapy. In contrast to the previous study, Nakamura et al. chose to use the psoas muscle index (PMI) as a marker for muscle mass and found that the strongest reduction in PMI occurs at the beginning of chemotherapy treatment. Additionally, younger children and boys tended to show a recovery in PMI between the second and last measurements [39]. Ijpma et al. measured the SMI at L3 on CT at diagnosis, after six cycles of chemotherapy, after autologous stem cell transplantation (ASCT), after three cycles of immunotherapy, and at the end of immunotherapy. In contrast to Nakamura et al. they found a minimal increase in skeletal muscle mass, skeletal muscle density, and intermuscular adipose tissue over time, with a rapid increase in visceral and subcutaneous adipose tissue after chemotherapy, suggesting that chemotherapy does not have much impact on muscle quantity and quality [129]. The influence of cancer treatment on muscle mass and how this change impacts outcome are two areas we are just beginning to understand. Due to the aggressive nature of high-risk NB, multimodal therapies and multiagent chemotherapy regimens are frequently used, making it difficult to isolate each factor influencing muscle mass. It is uncertain why the results of the above two studies differed, but it may be due to differences in patient ages, chemotherapy treatment (protocol JN-H-07 vs. DCOG NBL 2009 protocol) [233,234], the marker used, or the fact that muscle mass was already low before diagnoses in the later study.

Guo et al. used DXA to determine muscle mass in children who were in complete remission for at least two years after high-risk NB treatment. Compared with other studies in children with NB, Guo et al. assessed both muscle mass and muscle strength. The results showed that survivors had a lower leg lean muscle than age-, sex-, and race-matched controls. Muscle strength was measured using isometric ankle dynamometry, specifically dorsiflexion peak torque was calculated, and showed that survivors also had a significantly lower muscle strength [117]. Vatanen et al. also showed that long-term survivors of high-risk NB had a significantly lower LM. In addition, survivors also had a higher risk for a frail phenotype defined as having at least three of the following: low LM mass on DXA, low energy expenditure determined through an interview, slowness measured using the shuttle-run test, and weakness measured by the sit-up test [126]. These studies expand on our knowledge of sarcopenia in children with high-risk NB by showing that sarcopenia can be a persistent problem even years after cancer treatment.

In a cohort of children with NB at our center, we measured the tPMA at L3–4 and L4–5 on CT and MRI imagining prior to tumor surgery. While isolating the influences of chemotherapy was made more difficult by this approach, our results showed that most children with NB had a tPMA z-score below −2, which we defined as sarcopenic. Girls showed lower tPMA z-scores than boys, though they did not have a more progressive disease course. Causes could include a higher vulnerability to cancer treatment or tumor processes. Using sarcopenia to predict five-year mortality proved more useful in boys. Pre-operative sarcopenia, along with age at diagnosis, unfavorable tumor histology, and receiving NB2004-High Risk (HR) chemotherapy were risk factors of five-year mortality. This study showed that low tPMA z-score might be a useful marker for predicting mortality in children with NB [120].

Using similar methods, we also examined tPMA in children with hepatoblastoma (HB). Similar to children with NB, most children with HB were sarcopenic according to the above definition. Additionally, sarcopenia was a risk factor for relapse in children with high-risk HB. Due to high survival rates, the association between tPMA and mortality could not be determined. Similar to our findings in children with NB, girls with HB had a lower tPMA z-scores compared with boys, although they also did not have a less favorable disease status [119]. This difference was also observed in adults, which might be related to sex differences in muscle qualities (fibers) and/or inflammatory responses [235]. 

In 2021, body composition was also investigated in children with Ewing sarcoma, osteosarcoma, rhabdomyosarcoma, and Wilms tumor. Using cross-sectional CT images, Joffe et al. measured skeletal muscle, residual lean tissue, and adipose tissue at heights T12–L1 and L3. Overall, a decrease in skeletal muscle and residual lean tissue was found between diagnoses and after 6 to 12 weeks, with a significant increase in visceral adipose tissue. Older children also tended to have a higher loss in skeletal muscle [128]. Although the pathologies and the time of measurements vary, the research of Joffe et al. and Nakamura et al. may suggest that, in children with solid organ tumors, older children experience a higher loss in skeletal muscle, while younger children have a quicker recovery. 

Additionally, calculating the tPMA z-score on CT images at L4–5, but in children with Ewing sarcoma, rhabdomyosarcoma, and a desmoplastic tumor, Romano et al. found that the majority of patients were sarcopenic at diagnosis, as defined by tPMA z-scores below -1 (mild), -2 (moderate), or -3 (severe). One year after diagnosis, their tPMA z-scores continued to decrease significantly [130]. This study may give us further evidence that children with solid organ tumors commonly have low muscle mass, which may be further worsened by cancer treatment.

Numerous additional studies have been published examining body composition, specifically FM and FFM, for a wide variety of pediatric malignancies, including children with solid organ tumors. It is important to remember that FFM and SMM cannot be used interchangeably [236], although a recent study has found that FFM may predict SMM. In addition, FFM measurements using DXA and ADP may have a systemic error when measured in patients with obesity [237]. Below are some studies which have examined FM and FMM in children with solid organ tumors, possibly giving us more clues about SO in this population.

Brinksma et al. used BIA to determine body composition in kids with hematological malignancies, solid tumors, and brain tumors [238]. They found that, in the first year of diagnosis, FFM remained low while FM and triceps skinfold thickness increased. A rapid increase in the mid-upper arm circumference was also seen in the first three months [238]. Two published studies measured FFM and FM using ADP in a heterogeneous group of children with hematological and solid organ malignancies. During various stages of treatment, children had a significantly higher FM, while differences in FFM were lower but not significant [68]. In a second study with a larger group of patients, a reduced FFM was observed, while FM increased during cancer treatment. An increase in FM was also observed in cancer survivors [109]. Multiple studies have also been published using a deuterium dilution to measure FM and FFM. A recent publication determined no major difference when calculating FM and FFM between deuterium dilution and BIA [131]. One study using a deuterium dilution to predict FM and FFM in children with malignancies found no change in body composition in children with solid organ tumors six months after chemotherapy [122].

## 8. Conclusions and Future Outlook

With research on sarcopenia in children with cancer still in its infancy, many questions remain unanswered. Although many of the underlying pathophysiological mechanisms have been studied extensively in adults, only recently have more studies emerged examining the many causes of sarcopenia and cachexia in children, specifically those with cancer. Only by better understanding the underlying factors and pathophysiological processes causing sarcopenia can we better understand why sarcopenia may negatively impact outcome. 

Furthermore, the community must prioritize alignment on actionable definitions and standards for the identification of sarcopenia in children. Despite recent acceleration in pediatric sarcopenia research, individual studies are difficult to compare due to the lack of established definitions and diagnostic recommendations. Even within a specific cohort such as children with solid organ tumors, a wide range of modalities and protocols have been proposed to assess sarcopenia, with few incorporating direct measurements of strength and function. To conclude, a consensus on standardized protocols to measure muscle mass, strength, and function must be reached and guidelines to diagnose sarcopenia need to be established. Without these, it will remain difficult to translate research into effective clinical practice.

## Figures and Tables

**Figure 1 cells-11-01278-f001:**
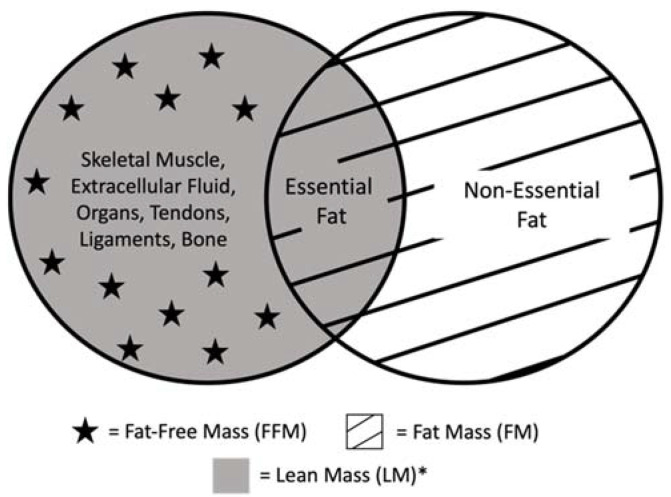
Breakdown of total body composition and definitions. Essential fat includes fat within the bone marrow and organs. Non-essential fat, or storage fat, includes fat surrounding organs and fat located subcutaneously. Within the research, the definitions of lean mass and fat-free mass frequently cause confusion. Some authors exclude bone in the definition of lean mass, and other authors equate lean mass with fat-free mass [9,10,11]. * lean mass (kg)/height (m^2^).

**Table 1 cells-11-01278-t001:** Factors that may contribute to sarcopenia in children with solid organ tumors.

Direct Factors	Indirect Factors
Reduced physical activity Medications − Chemotherapy − Doxorubicin − Cisplatin − Steroid use Radiation Cancer cachexia − Tumor inflammation − Organ dysfunction − Pain/stress Endocrine − Peptide ghrelin − Vitamin D	Fatigue Chemotherapy − Mucositis − Nausea and Vomiting − Diarrhea

**Table 2 cells-11-01278-t002:** Summary of current research on sarcopenia in children with solid organ tumors.

Author, Year	Cohort Size	Tumor Type	Age (Years)	Measurement Modality	Markers for Muscle Mass	Markers for Muscle Strength or Function	Summary
Vatanen et al., 2017 [126]	*n* = 19	High-risk NB	median 22 (range 16–30)	DXA	Whole body LMI *	Strength: Sit-up test Function: Shuttle-run test ^§^	Survivors have a low LM and higher risk of frail health
Kawakubo et al., 2019 [127]	*n* = 13	High-risk NB	PFS: mean 2 (range 0–5) R/D: mean 3.1 (range 2–5)	CT	tPMA at L3		During standard treatment, the rate of change increased in the progression-free survival group and decreased in the relapse and death group
Joffe et al., 2020 [128]	*N* = 39 (*n* = 8, 7, 16, 8, respectively)	Ewing sarcoma, Osteosarcoma, Rhabdomyosarcoma, Wilms tumor	median 11 (range 1.33–20)	CT	SMM at T12-L1 (*n* = 39), L3 (*n* = 22)		After 6–12 weeks, skeletal muscle and residual lean tissue decreased and visceral adipose tissue increased
Nakamura et al., 2021 [39]	*n* = 24	High-risk NB	median 2 (range 0–6)	CT	PMI ^†^ at L4		During induction chemotherapy, the strongest reduction in PMI occurred at the beginning of chemotherapy and younger children and boys tended to show a recovery in PMI between the second and last measurements
Guo et al., 2021 [117]	*n* = 20	High-risk NB	mean 12.4 ± SD 1.6	DXA	Leg LM, appendicular LM, total body LM (excluding bone mass)	Strength: isometric ankle dynamometry	Survivors have a low leg LM and muscle strength
Ritz et al., 2021 [120]	*n* = 101	NB	median 3 (IQR 2.25–5)	CT, MRI	tPMA at L3–4, 4–5		Before surgery, the majority had tPMA z < −2 and pre-operative tPMA z < −2 risk factor for 5-year mortality, and girls have lower tPMA z-scores
Ijpma et al., 2021 [129]	*n* = 29	High-risk NB	median 3.0 (IQR 2.0–4.5)	CT	SMI ^‡^, skeletal muscle density at L3		During treatment, skeletal muscle mass, skeletal muscle density, and intermuscular adipose tissue increased minimally and visceral and subcutaneous adipose tissue increased
Ritz et al., 2021 [119]	*n* = 33	Hepatoblastoma	median 2.15 (IQR 1.47, 3.24)	CT, MRI	tPMA at L3–4, 4–5		Before surgery, majority had tPMA z < −2; in high-risk HB, pre-operative tPMA z < −2 was a risk factor for relapse; and girls have lower tPMA z-scores
Romano et al., 2022 [130]	*n* = 21	Ewing Sarcoma, Rhabdomyosarcoma, Desmoplastic tumor	median 10.5 (IQR 6.6, 15.1)	CT	tPMA at L4–5		At diagnosis and after 1 year, majority had tPMA z < −1 and tPMA z decreased significantly after 1 year

* lean mass (kg)/height (m^2^). ^†^ psoas muscle cross-sectional area (PMCSA, cm^2^)/body surface area (m^2^) = (PMCSA, cm^2^)/(square root of (height [cm] × weight [kg]/3600)). ^‡^ skeletal muscle cross-sectional area (cm^2^)/height (m^2^). ^§^ The purpose of this study was to assess for frailty, not sarcopenia. The shuttle-run test was used as a marker for slowness. The shuttle-run test could also be interpreted as a marker for muscle function (performance). Abbreviations: IQR, interquartile range; LM, lean mass; LMI, lean mass index; NB, neuroblastoma; PFS, progression-free survival; PMI, psoas muscle index; R/D, relapse/death; SD, standard deviation; SMM, skeletal muscle mass; SMI, skeletal muscle index; MUAC, mid-upper arm circumference; z, z-score

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
