# Peer review of "Sarcopenia in Children with Solid Organ Tumors: An Instrumental Era"

_cells, 2022, doi:10.3390/cells11081278_

Round 1

Reviewer 1 Report

I believe that the work contains interesting contents about the problem of sarcopenia in pediatric patients with solid tumor, but it is necessary to improve the English and the form in which the contents are expressed.

Reviewer 2 Report

In this paper Ritz et al present an interesting and detailed review on sarcopenia and malnutrition in children with solid organ tumor. Structure of the paper is well organized. The topic is analyzed criticality even if literature about it lacks and some points are to be better specified.

A number of both major and minor issues should be addressed:

MAJOR ISSUES

  • The title of the review highlights the role of biomarkers, that in the paper are not cited in relation to pediatric population but only in the general part (about adults). Which role do they have in children? Are there studies about it? Otherwise a change in the title could be less confounding. Even in the table, which is the “core” of the article, data proposed are all instrumental.
  • The article is detailed and comprehensive of many issues. Probably this may lead to pay less attention to the main topic of this review. For this reason I suggest to delete paragraph 6, adding what is essential of it as an introduction to the next paragraph. Furthermore, paragraphs 7 and 8 could be joined in order to highlight the most important aspect of the review which is precisely the sarcopenia in children with solid tumors.
  • A specification about children ages may be needed. In fact, body compostions are widely different among childrens of varous age group and for this reason, some considerations about sarcopenia must be separetely analyzed. This aspects is only briefly explained and needs more emphasis.
  • Given the complexity and the numerosity of factors that can lead to sarcopenia in cancer patients, it might be useful to summarize them in a table or figure, dividing them by direct and indirect effects. Their understanding is fundamental and a visual summary is needed.
  • The definition of sarcopenic obesity at the beginnig of paragraph 3 is confusing. It is generally defined as a loss of muscle mass and strenght (sarcopenia) together with the gain of adipose tissue (obesity) rather than the infiltration of fat into the muscle. Furthermore, the definition proposed in the paragraph is not supported by a literature citation.
  • In the paragraph 4, frialty is described. The authors suggest that to overcome the limit of frailty assestment in childrens younger that 5 years, imaging of psoas muscle may be used. This approach is useful, but in the context of sarcopenia and not in the context of frailty, which are two differt enties. In the adult patiets frailty does correlate to sarcopenia, but it is a global performance-related concept that needs also functional examination. 

MINOR ISSUES

  • Lack of citations: many concepts are reported in the paper but without specific references. Examples are lines 59, 94-97, 140-142, 159, 163, 173-177, 246, 290.
  • Some abbrevations are reported without extensive name (or the name is explain only after). Examples: ALL, SMA, FFM, FM. Moreover, in line 178, C57/BL6 is reported without explanation.
  • An explication about the choice to speak also of the ALL must be added since it is not a solid tumor.
  • At line 313-323 electronic system of muscle muss analyis are named. This may be unnecessary since there are no studies related to childrens and removing them could lighten the reading.
  • In the conclutions the authors propose many questions that remains open. My suggestion is to convert them in more concrete studies proposals, in order to make the conclusions and the paper useful for establish future reserch agenda.

Reviewer 3 Report

The authors reviewed sarcopenia in children with solid organ tumors. They did not, however, define the age of the children in their review. Even when children share the same pathological conditions, there are large variations in the development status of their bodies, including in the skeletal muscle mass and function. Further, almost all of descriptions in the review are based on publications covering adults. The effects of disuse and medications on protein balance in skeletal muscle in children remain unclear. The endocrine status in children may differ from that in adults. The authors should focus on the pathological differences and diagnostic points of sarcopenia in children compared to adults.

Regarding the biomarker, almost all of the descriptions are based on the total psoas muscle area (tPMA). Is tPMA a biomarker for sarcopenia in children? 

As the authors mentioned, sarcopenia is a progressive loss of skeletal muscle mass. During development, on the other hand, the skeletal muscle mass and function increase. As tumors have progressive pathologies, the atrophic status of the skeletal muscle depends on the pathologic stage of the tumors. Skeletal muscle mass is determined by complicated factors. Can the skeletal muscle mass be considered a biomarker for sarcopenia with tumors under such conditions?

Line 59: “exercise decreased protein degradation.”

Is this true? If true, the evidence should be shown.

Line 60: “Physical exercise also reduces reactive oxygen species (ROS)”

Is this true? If true, the evidence should be shown.

Line 52: “ADP”

“ADP” should be spelled out.

Reviewer 4 Report

A couple of points to emphasize in the review will enhance the comprehensive value of this review to the literature.

  1. a) There is evidence that DEXA may under-estimate sarcopenia in adults and children. However, more recent data related the use of cross-sectional /segmental DEXA or BIA has improved ability to measure sarcopenia in children (Belarmino et al World J Hepatol 2017). 
  2. b) MR control data to assess SMM and adiposity in healthy children is limited. While the concept about MRI being used interchangeably with CT for determination of SMM has been shown in adults, there is limited information regarding the ability to assess subcutaneous and visceral adiposity using MR and CT interchangeably.  This is particularly relevant in childhood where rapid changes in SAT and VAT can occur with growth and under and over nutrition.  It would be beneficial to highlight this particular point as a potential limitation in the pediatric sarcopenic and cancer literature. This is particularly relevant in children post BMT, where data has indicated (as per the authors) of relative increases in adiposity in the overall context of sarcopenia.
  3. c) Use of tPMA z-scores to determine cut-offs for sarcopenia in childhood has several potential limitations, particularly in reference to alternations in growth patterns due to suboptimal nutrition and differences between sexes throughout childhood. While the authors acknowledge some of the limitations, it would be important to identify whether these limitations have the potential to result in over or underestimations of sarcopenia prevalence in childhood. This would be important in highlighting the need for additional research in this important area.

Round 2

Reviewer 3 Report

The authors revised their manuscript in response to an earlier round of reviewer comments. Their revisions, however, are incomplete.

In my previous review, I pointed out that it remains unclear how muscle disuse and medications affect the protein balance in the skeletal muscle of children. The capacity stimulated in the skeletal muscle of children may be more anabolic than proteolytic. In adult skeletal muscle, on the other hand, the protein balance is stable. The manuscript does not address this important issue.